# Numerical Determination of RVE for Heterogeneous Geomaterials Based on Digital Image Processing Technology

**Lanlan Yang [1], Weiya Xu [1,\*], Qingxiang Meng [1,2], Wei-Chau Xie [1,3], Huanling Wang [1,4] and Mengcheng Sun [1]**

[1]  Research Institute of Geotechnical Engineering, Hohai University, Nanjing 210098, China; yanglanlan_hhu@163.com (L.Y.); mqx4088@gmail.com (Q.M.); xie@uwaterloo.ca (W.-C.X.); whl_hm@163.com (H.W.); mc-sun@hhu.edu.cn (M.S.)
[2]  College of Water Conservancy and Hydropower Engineering, Hohai University, Nanjing 210098, China
[3]  Department of Civil and Environmental Engineering, University of Waterloo, Waterloo, Ontario N2L 3G1, Canada
[4]  Key Laboratory of Coastal Disaster and Defense, Ministry of Education, Hohai University, Nanjing 210098, China
\*  Correspondence: wyxuhhu@163.com; Tel.: +86-025-8378-7037

**Abstract:** Representative volume element (RVE) is an important parameter in numerical tests of mechanical properties of heterogeneous geomaterials. For this study, a digital image processing (DIP) technology was proposed for estimating the RVE of heterogeneous geomaterials. A color image of soil and rock mixture (SRM) with size of $400 \times 400$ mm$^2$ taken from a large landslide was used to illustrate the determination procedure of the SRM. Six sample sizes ranging from $40 \times 40$ mm$^2$ to $240 \times 240$ mm$^2$ were investigated, and twelve random samples were taken from the binarized image for each sample size. A connected-component labeling algorithm was introduced to identify the microstructure. After establishing the numerical finite difference models of the samples, a set of numerical triaxial tests under different confining pressures were carried out. Results show that the size of SRM sample affects the estimation of the mechanical properties, including compressive strength, cohesion, and internal friction angle. The larger the size of the samples, the less variability of the estimated mechanical properties. The coefficient of variation (CV) was applied to measure the variability of mechanical properties, and the RVE of the SRM was determined easily with a predefined acceptance threshold of the CV. The results show that a DIP-based modeling method is an effective method got the RVE determination of heterogeneous geomaterials.

**Keywords:** heterogeneous geomaterials; digital image processing; DIP; representative volume element; RVE; mechanical properties; coefficient of variation; CV

## 1. Introduction

Soil–rock mixture (SRM), as a multiphase geomaterial composed of many components such as rocks, soils, and pores, is a special geological body often encountered in geotechnical engineering [1]. Individual ingredients may have different physical and mechanical properties (such as density, porosity, and compressive strength) and different distribution patterns. As a complex heterogeneous material, the macroscopic physical and mechanical properties of the SRM (such as strength parameters, failure modes, and constitutive equation) rely largely on the intrinsic mesostructures, including particle shape, distribution, and proportion of each ingredient. It has been well recognized that inhomogeneities and mesostructures are of great significance to the physical and mechanical properties of geomaterials [2].



In order to investigate the effects of inhomogeneities and microstructures, many efforts have been made in the numerical simulation of geomaterials. It is known that the size of samples is of great importance when inhomogeneous material behavior is modelled and studied. A multiphase material can be regarded as a homogeneous medium when the size of sample is large enough, with sufficient mesoheterogeneities (such as voids, cracks, rocks, etc.) included [3]. As a result, the concept of representative volume element (RVE) is of critical significance in the numerical modelling and simulation of heterogeneous material. Although a general expression for RVE of an arbitrary heterogeneous material does not exist, different definitions of the RVE have been adopted for different materials, properties and purposes [4]. Simply speaking, a sample of heterogeneous material is considered as a RVE when (i) the investigated homogenized properties do not change to a large degree with the increase of sample size and (ii) its size is the minimum that satisfies the requirement of statistical homogeneity [5]. Several methods have been proposed for RVE determination of heterogeneous materials. Generally, these procedures can be categorized as a constitutive equation-based quantitative estimation method [6–10] and a numerical simulation method. With respect to different modeling approaches, there are three options of numerical determination methods, namely, circular and spherical inclusion models [3–5,11,12] and fractured systems [13–16].

Although these approaches perform well in investigating the statistical characteristics and scale effect of composite materials and simplified concrete materials, it is hard to describe the real shape, orientation, and distribution of rocks in soil and rock mixture (SRM) for a specific site which may have influence on the mechanical parameters and failure mode as well. The widely used digital image processing (DIP) is capable of capturing morphological characteristics in engineering, information science, statistics, physics, and other disciplines [17–26]. Particularly, Kameda established a digital rock approach for the analysis of permeability evolution in sandstone [26]. Further enhancement was made by incorporating digital image processing techniques into the numerical method [1,27–35]. Yue et al. proposed a quantitative analysis procedure entitled DIP-based finite element method (DIP-FEM) for asphalt concrete considering the inhomogeneities [34]. Meng et al. incorporated DIP with the discrete element method (DIP-DEM) for the analysis of inhomogeneous media [36]. Zhang et al. reconstructed stochastic elliptical surrogate models according to the statistical characteristics extracted by DIP from a color SRM photo and investigated the scale effect on the unconfined compressive strength [37].

The DIP technique is an effective tool in quantitative analysis, as it can provide an explicit representation of heterogeneity and microstructure of a geomaterial [38]. For the merits of DIP, the main purpose of this study was to propose a novel method for the determination of representative volume element. The general flowchart of the main techniques and implementation procedures is shown in Figure 1, and the presentation of this paper is arranged as follows:

1. The pre-processing technique for digital images and the method to generate samples for analysis are introduced briefly.
2. A DIP method for geomaterials, based on a connected-component labeling algorithm, is then presented. An implementation of this method to convert a digital image from an original figure to a vectorized drawing is presented.
3. An algorithm for generating a 2D finite difference element meshes automatically from vectorized images is applied. The 2D numerical grid is converted to a 3D numerical model through a proposed interface program.
4. The scale effect of macro-mechanical properties is investigated by conducting numerical triaxial compressive tests. A method for estimating the REV of an SRM is proposed.

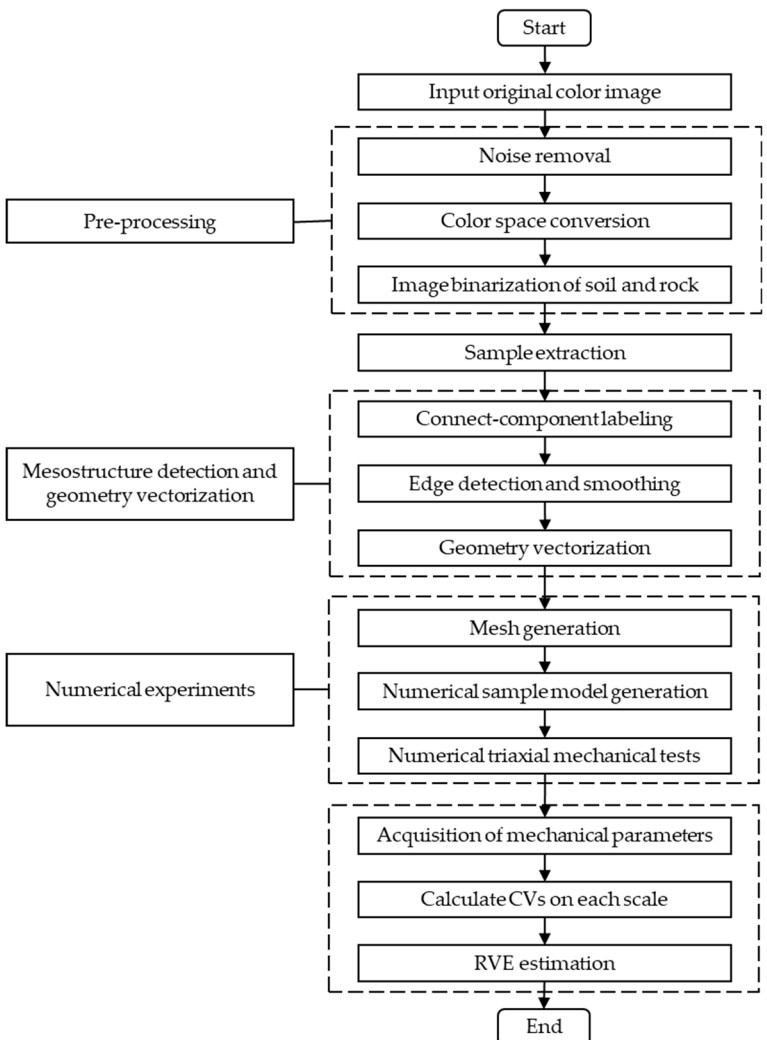

**Figure 1.** Flowchart for representative volume element (RVE) estimation of soil and rock mixture (SRM) based on digital image processing (DIP) technology.

## 2. Numerical Sample Generation

### 2.1. Pre-Processing of Digital Image

With the popularization of digital cameras, images of geomaterial surfaces can be easily digitized and stored in digital graphic files. A digital photograph captured by a digital camera or a scanner consists of a rectangular matrix of image elements or pixels. Formats of gray images and color images are different. Each pixel is represented by an integer value from 0 to 255 for gray images, and from 0 to 1 for binary images. Hence, digital images contenting $m \times n$ pixels can be represented by a discrete function $f(x, y)$ in the $x$ and $y$ Cartesian coordinate system [34]:

$$f(x, y) = \begin{bmatrix} f(1,1) & f(1,2) & \cdots & f(1,m) \\ f(2,1) & f(2,2) & \cdots & f(2,m) \\ \vdots & \vdots & \ddots & \vdots \\ f(n,1) & f(n,2) & \cdots & f(n,m) \end{bmatrix}, \tag{1}$$

where $x$ ranges from 1 to $n$ and $y$ ranges from 1 to $m$.

For color images, three integer values are employed to represent different color components, such as red, green, and blue. As a result, the data of color images can be expressed as a function $f_k(x, y)$,

where $k$ = 1, 2, 3. In this paper, a photo of soil–rock mixture (SRM), as shown in Figure 2, is used as an example. This digital image has $N_x = N_y$ = 2000 pixels in the x- and y-directions, respectively. According to the actual scale of the geomaterial, each pixel in this photo has an actual length of 0.2 mm, that is, $L_x = L_y$ = 400 mm.

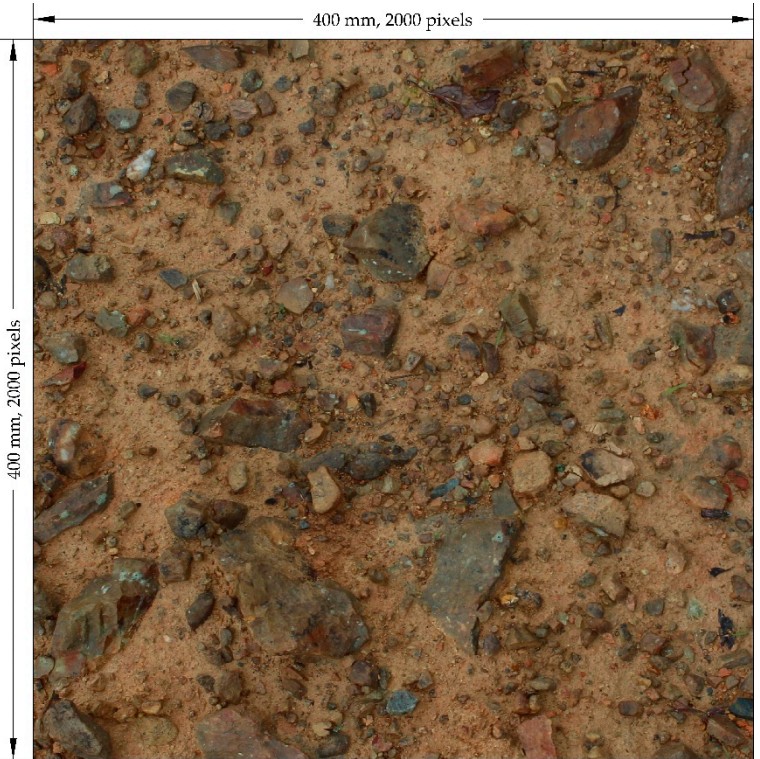

**Figure 2.** A color photograph of soil and rock mixtures from a landslide project.

### 2.1.1. Noise Removal

Noise is inevitable in the acquisition of digital images. There are many sources that could introduce noise, which can be divided into two categories: environmental factors and internal factors. These noises affect the extraction of microstructural information of geomaterials. Hence, noise elimination is essential in image pre-processing. An easy yet effective method, called the median method, which is also used in [39], is employed in this paper.

Median filtering is a type of nonlinear smoothing method that keeps the boundaries of original figures clear. Furthermore, neighboring windows have a variety of shapes, such as a circle, rectangle, and cross. A square with a size of 3 or 5 pixels is mostly used in practice. The function of the median filter [36] can be written easily as follows:

$$g(x, y) = \text{median}\{f(x - K, y - I), (K, I \in W)\}, \tag{2}$$

where $W$ is the neighboring window. In this paper, a simple neighboring window of a square with a size of 3 pixels was used. Programs for the median filter and other related programs were implemented in MATLAB (Version 2017b, Waterloo, ON, Canada).

### 2.1.2. Color Space Conversion

Digital images produced by most scanners and cameras are color images which are characterized by certain color spaces. Since all colors can be regarded as a combination of red (R), green (G), and blue (B), the RGB color space is now the most frequently used. Despite the fact that the RGB color system is accepted by human eyes, it is not effective for computers to recognize objects from digital images.

Other color spaces, such as YUV (including the components of Luminance, Chrominance, and Chroma), YIQ (stands for Luminance, In-phase, and Quadrature-phase), CMYK (shorts for Cyan, Magenta, Yellow, and Black), and HSI (consisting of Hue, Saturation, and Intensity), are also available. In this paper, original images were captured by cameras in the RGB color system, but the digital images were analyzed in the HSI color space, which is more convenient in color manipulation and better at object recognition.

The HSI color space can be described by a circular cone model, as shown in Figure 3. Although the expression of the HSI color space is very complicated, it is a more accurate and has a more distinct description of a color image. For the convenience of digital processing, images in RGB color space are converted to HSI color space. The essence of this process is the transformation of a unit cube in Cartesian coordinates to a bipyramid in cylindrical coordinates. A simple transformation [36] is presented as follows.

$$H = \begin{cases} \theta, & G \geq B \\ 2\pi - \theta, & G < B \end{cases}; \quad S = 1 - \frac{3\min(R, G, B)}{R + G + B}; \quad I = \frac{1}{3}(R + G + B), \tag{3}$$

in which $\theta = \cos^{-1}\left[\frac{(R-G)+(R-B)}{2\sqrt{(R-G)^2+(R-G)(R-B)}}\right]$. Other methods, such as coordinate transformation [40], Bajon approximation [41], and piecewise definition [42], can also be applied.

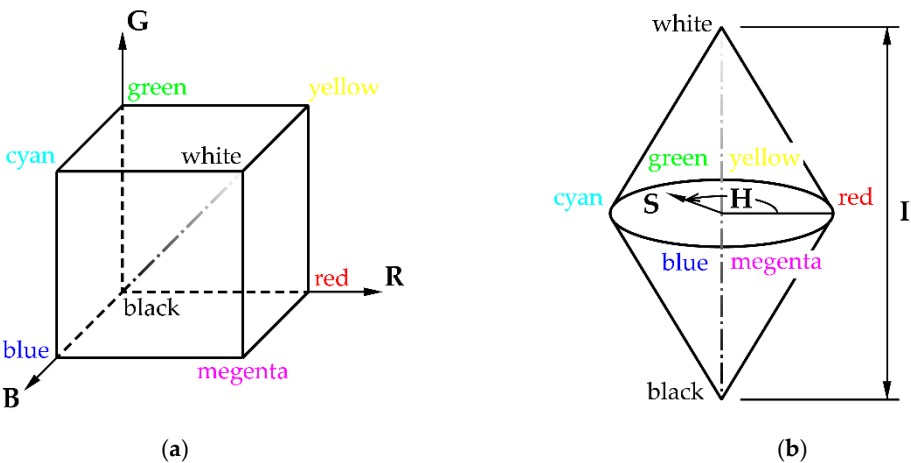

(**a**)            (**b**)

**Figure 3.** The RGB (red, green, blue) and HSI (Hue, Saturation, and Intensity) color space illustration: (**a**) RGB color space; (**b**) HSI color space [36].

A MATLAB program RGB2HSI, provided in Gonzalez et al. [43], was employed in the digital analysis of geomaterial images. It was found that saturation could better distinguish components than hue and intensity. Therefore, rocks in images can be distinguished based on saturation.

### 2.1.3. Image Binarization

According to the discussion above, the grayscale of saturation (S) was taken as the differentiating factor between rocks and soil. The determination of a proper threshold is very important. As per the histogram of saturation (S) value shown in Figure 4, the value of saturation increases before 0.2891 and falls gradually after the peak. Therefore, a threshold value of $S_{cr} = 0.2891$ was selected for the binarization. Since the distinction of soil and rocks is not evident enough and rock surfaces are sometimes covered by soil, manual refinement was needed. The obtained binary image of the SRM microstructure is shown in Figure 5, in which the rock aggregates can be easily identified.

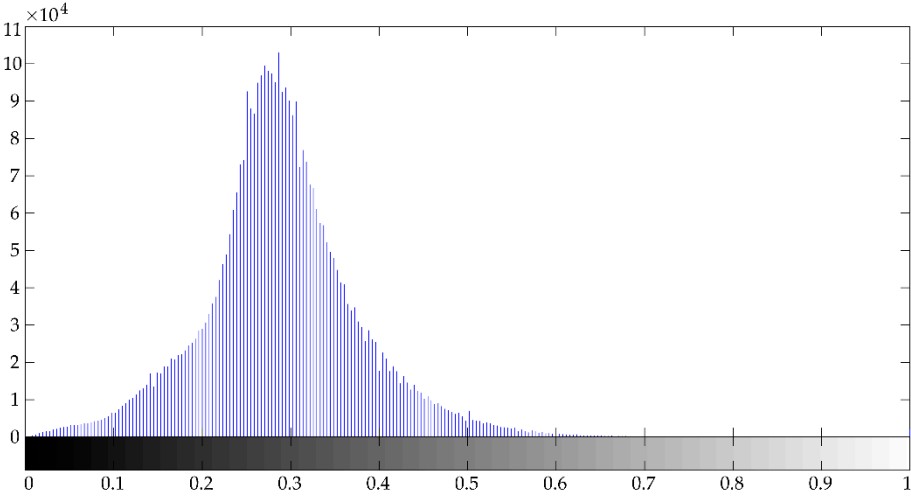

**Figure 4.** Histogram of saturation (S) value.

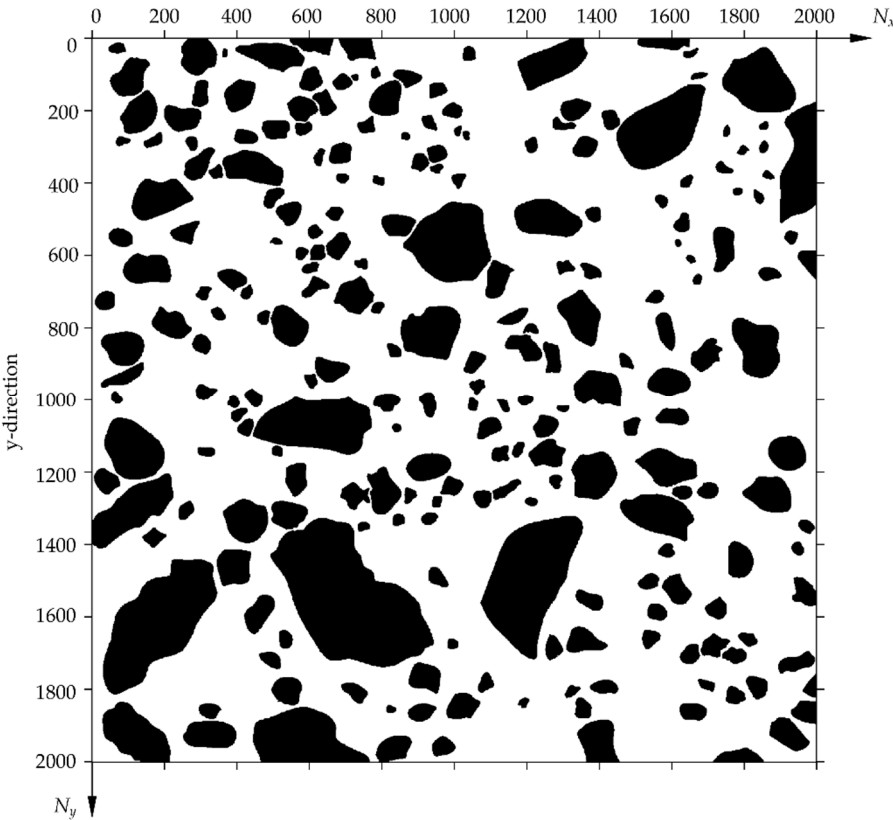

**Figure 5.** Final binary image of the SRM picture.

### 2.1.4. Connected-Component Labeling Algorithm

As per the binary image shown in Figure 5, each aggregate is formed by all adjacent pixels. Hence, each rock aggregate can be viewed as a connected component. As a result, the connected-component labeling algorithm can be applied to tackle this issue.

A connected-component labeling algorithm is generally used to locate and count the aggregates by labeling the connected pixels of a binary image. At first, a unique number of 1 is assigned to each target blob, whereas the number of 0 is assigned to each background blob. If each target blob corresponds to a single object, these objects in the binary image can be counted and labeled. Generally speaking, the labeling algorithm can be divided into two large classes: (a) local neighborhood

algorithms and (b) divide-and-conquer algorithms. For the local neighborhood algorithms, 4- or 8-connectivity in component labeling is often used. Since there are at most 8 pixels adjacent to a single pixel, a 4-connectivity algorithm identifies the pixels with at least one coinciding side, whereas an 8-connectivity algorithm identifies all adjacent pixels as a component (Figure 6). For the identification of different rocks, an 8-connectivity labeling algorithm is used, and the distribution of aggregates can be determined based on the results of labeling.

|   |   |   |   |   |   |
|---|---|---|---|---|---|
| 1 | 0 | 1 | 1 | 0 | 0 |
| 1 | 0 | 1 | 1 | 0 | 0 |
| 1 | 0 | 0 | 0 | 1 | 0 |
| 1 | 1 | 0 | 0 | 1 | 0 |
| 1 | 0 | 0 | 1 | 1 | 0 |
| 1 | 0 | 0 | 0 | 0 | 0 |

(a)

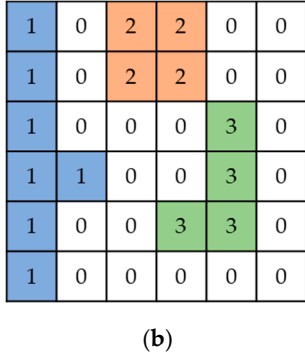

(b)

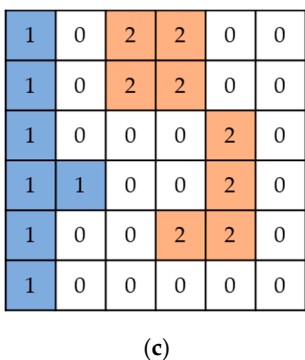

(c)

**Figure 6.** Illustration of connected-component labeling algorithm: (**a**) binary image; (**b**) three aggregates are detected by 4-connectivity component labeling; (**c**) two aggregates are detected by 8-connectivity component labeling.

As a result, statistical information of the rock size by using connected-component labeling algorithm is obtained by a program based on this procedure. The rock content (volume fraction) is $\rho$ = 30.54% in the binary image of the SRM from the statistical analysis. The result shows that the connected-component labeling algorithm is efficient in the analysis of the distribution of SRM components.

*2.2. Sample Extraction*

In order to estimate the size of the RVE for the SRM shown in Figure 2, sample pictures with different sizes had to be obtained, because of the scale effect of the SRM [4]. In this paper, the samples were generated from the binary image of the SRM obtained following the procedure in Section 2.1. There are several methods for generating samples, which can be classified into two categories. One option is to generate the samples randomly with the pre-set properties such as sample size, particle grading, and density $\rho$ (volume fraction) of each phase using certain programs as in [3,7]. The other option is to cut square domains from the binary image [1].

As an easy yet effective method, the second method was applied to extract a series of sample pictures randomly from the original digital image. Six sample sizes varying from $N_x = N_y = 200$ pixels to $N_x = N_y = 1200$ pixels, with actual sizes varying from $L_x = L_y = 40$ mm to $L_x = L_y = 240$ mm, were studied. Considering that the influence of rock content cannot be ignored [44], the samples were selected with the same volume fraction as the original binary image with a tolerance of ±1.00%, that is, the acceptable rock content range being $[\rho - 1.00\%, \rho + 1.00\%] = [29.54\%, 31.54\%]$, for a fair study. Randomness of rock distribution was considered by selecting 12 samples randomly, as shown in Figure 7 for example, for each sample size.

As shown in Figure 8, the centers of samples cut from the original binary image were distributed randomly for different sample sizes. It should be noted that 12 is a small number for a statistical analysis. Instead of providing a full statistical analysis, the purpose of this part is to illustrate the statistical background and procedure.

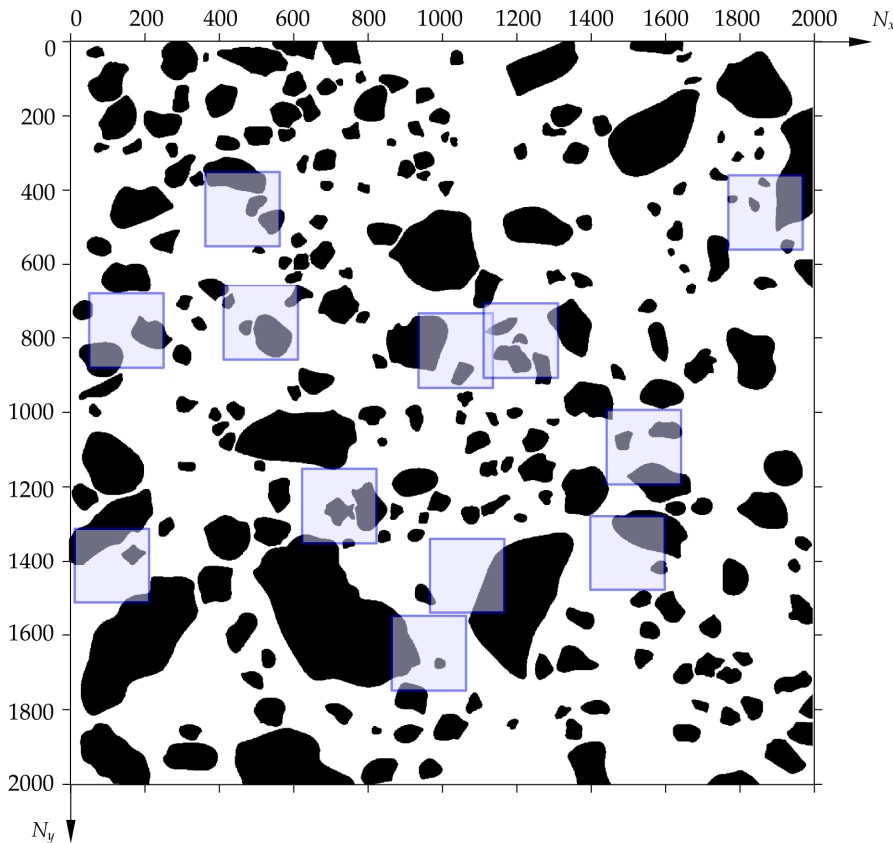

**Figure 7.** Samples with size of 200 × 200 pixels extracted randomly from a binary image of the SRM.

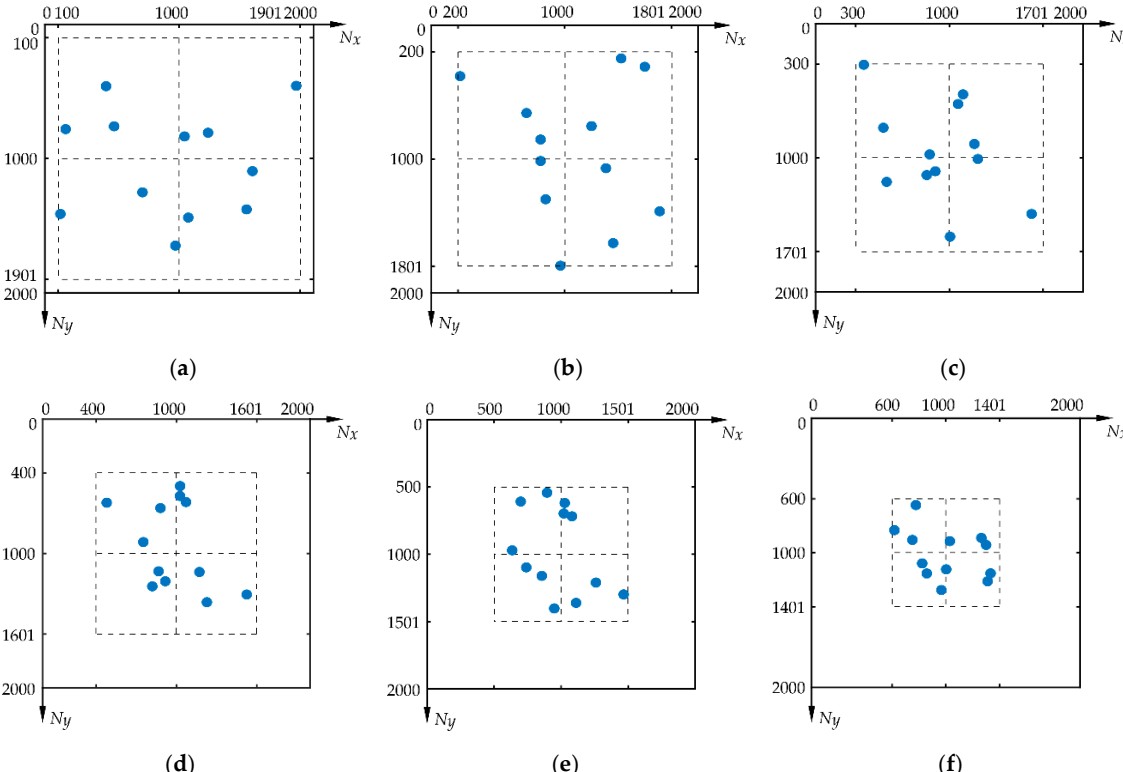

**Figure 8.** Distribution of sample centers extracted from a binary image of the SRM at different size levels: (**a**) 200 × 200 pixels; (**b**) 400 × 400 pixels; (**c**) 600 × 600 pixels; (**d**) 800 × 800 pixels; (**e**) 1000 × 1000 pixels; (**f**) 1200 × 1200 pixels.

*2.3. Microstructure Detection and Geometry Vectorization of Binary Image*

　　　Edge detection is an essential procedure in the vectorization of binary images. One pixel is a square with four sides. A connected-component of $N$ pixels has $4 \times N$ sides in total. If a side is at the edge of the domain, the side will be unique, while all other sides will repeat twice. If all the repeated sides are deleted, the remainder is the scattered edges of the connected-component. Having obtained the scattered edges, the next step is to connect the edges together in a sequential order. A computer program was developed following this procedure to extract the edge of rocks.

　　　After extracting the edge, the boundary of the rock was jagged, which could not reflect the real edge of the rock. Furthermore, an excessive number of nodes negatively affects the generation of computational models. Hence, a boundary processing method presented by Yue et al. [34] was applied. The method, illustrated in Figure 9, is briefly introduced as follows:

1.　Set a threshold value $T$.
2.　Find two points with the longest distance, mark them, and link them to divide the polyline into two segments.
3.　For any part, calculate the maximum perpendicular distance among all points in this part. If the maximum distance of this part is longer than the threshold value $T$, separate this part into two segments. Mark the corresponding point and link it to the former two marked points.
4.　Repeat step 3 until the perpendicular distance of all the segments is less than $T$.
5.　Connect all marked points one by one to form a polygon.

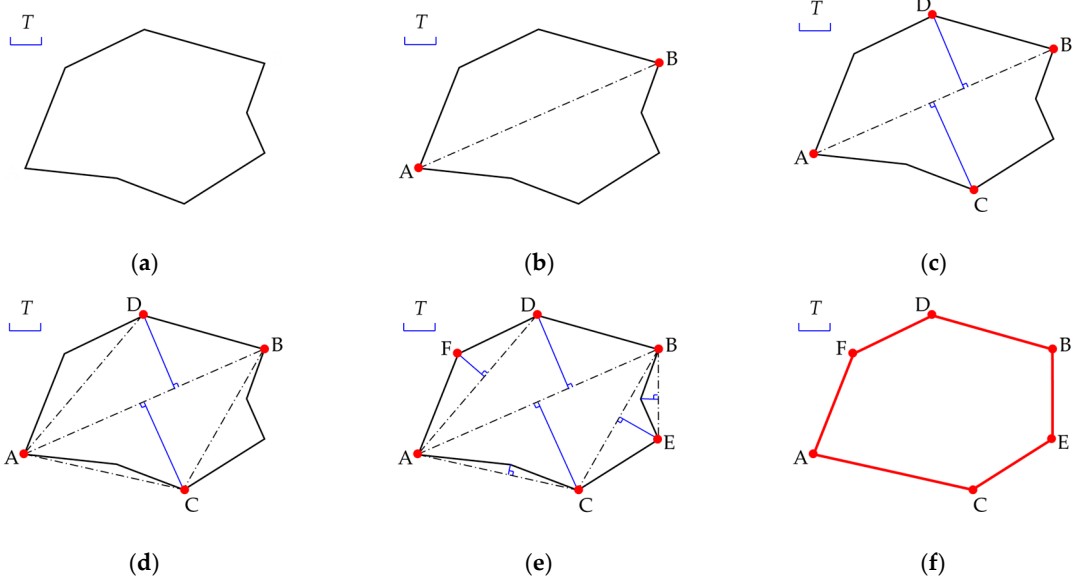

**Figure 9.** Illustration of boundary smoothing procedure: (**a**) step 1; (**b**) step 2; (**c**,**d**) step 3; (**e**) step 4; (**f**) step 5.

　　　The result for the threshold value $T = 10$ pixels is shown in Figure 10; it can be clearly seen that the points of the rocks are reduced. After obtaining the boundaries, a scale transformation is necessary. Since the actual size corresponding to each pixel is known, the position of each point and the real size of the SRM samples can be obtained by multiplying by the actual length of each pixel.

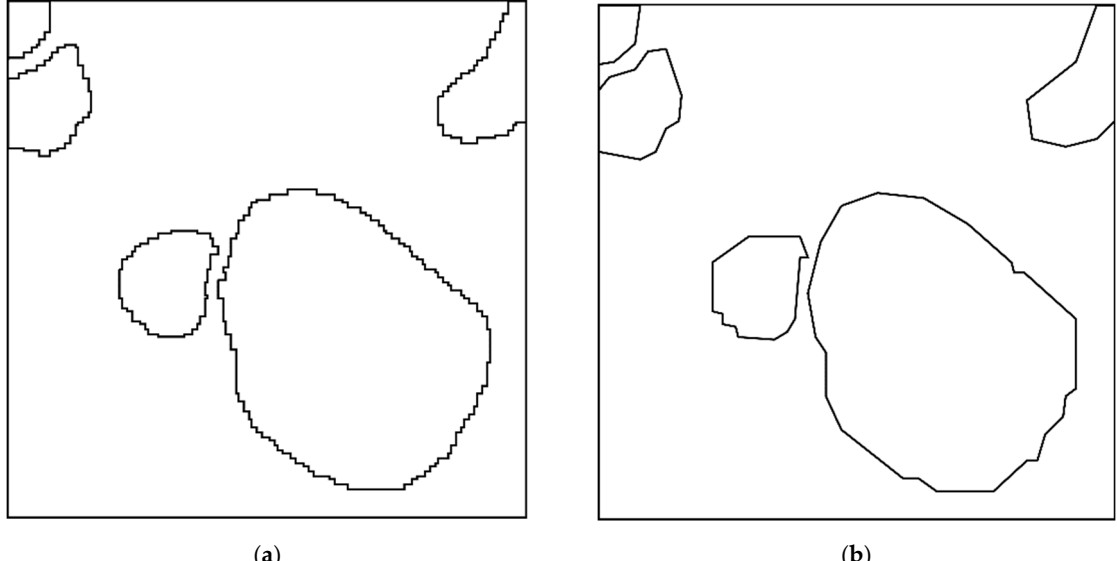

(**a**)　　　　　　　　　　　　　　　　　　　　　(**b**)

**Figure 10.** Edge detection and refinement of rocks in a sample of $200 \times 200$ pixels: (**a**) jagged rock edges; (**b**) refined rock edges.

## 3. Numerical Triaxial Mechanical Tests

### 3.1. Mesh Generation

The automatic generation of grids for numerical analysis is a classical topic accompanied by the boom of numerical methods. There are many techniques for the generation of finite element grids; three classical techniques, namely, quadtree-octree-based method, advancing front method, and Delaunay-based method, are the most widely used. Currently, the development of the FEM, such as arbitrary Lagrangian–Eulerian (ALE), brings new mesh requirements, such as adaptation and moving techniques. In this paper, requirements for grids are simple and conventional techniques are sufficient.

Many commercial and open source software packages, such as Ansys, HyperWorks, and Gmsh, can mesh finite element grids automatically. By importing vectorized image files into such a software, a desired mesh can be easily generated. For the convenience of later analysis, Gmsh software (Version 2.12.0) a fast, light, and user-friendly interactive mesh tool, was employed to generate mesh grids. As a result, the geometry information including mesh size, node coordinates, line segments, and line loops should be included in a *.GEO file, which is the recommended input file format. A program was developed to write the geometry information in the *.GEO file format.

By importing the *.GEO files of the SRM samples obtained in Section 2.2, 2D finite element meshes were generated using Gmsh. A mesh example for a sample with size of $40 \times 40$ mm$^2$ is shown in Figure 11a, of which the meshes are stored in *.MSH file format containing the physical group names ("rock" and "soil"), the nodes, and the elements (the element type of 3-node triangle is used for all samples).

In the later numerical study, a finite difference method (FDM)-based software Flac3D (Version 3.0, Nanjing, China) was applied to carry out numerical triaxial mechanical tests of the SRM for estimating the size of the RVE. An interface program Gmsh2Flac was written to map the 2D meshes (3-node triangle elements) to the 3D meshes (6-node wedge elements) along the z-direction with a certain thickness (Figure 11b). The 3D meshes were stored in a *.FLAC3D file which was acceptable as a numerical model in Flac3D. As a result, a total of 72 pseudo three-dimensional finite element models was obtained.

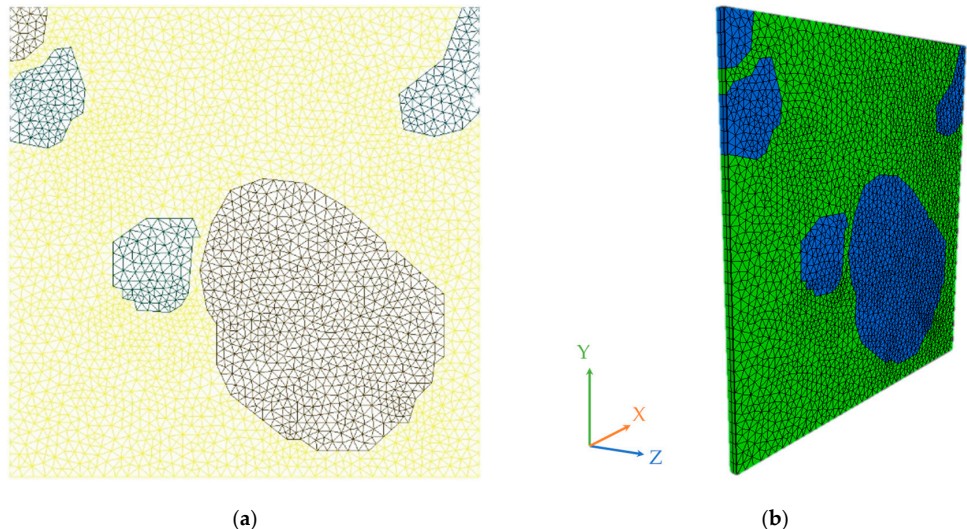

(**a**)　　　　　　　　　　　　　　　　　　　　　　(**b**)

**Figure 11.** Example of the SRM of size $40 \times 40$ mm$^2$: (**a**) mesh information stored in the *.MSH file; (**b**) finite difference numerical model.

### 3.2. Mechanical Parameters and Boundary Condition

Triaxial testing is the most commonly used method to study deformations and mechanical properties in geotechnical engineering [45]. To study the scale effect of mechanical properties of the SRM, a series of numerical pseudo triaxial compressive tests were conducted. The mechanical properties of different components of the SRM obtained from a landslide are listed in Table 1. It is clear that rock aggregates and soil have different mechanical properties.

**Table 1.** Mechanical properties for soil and rock components.

|  | Density $\rho$ (kg/m$^3$) | Bulk Modulus $K$ (GPa) | Shear Modulus $G$ (GPa) | Cohesion $c$ (kPa) | Friction Angle $\varphi$ (°) | Tension Strength $\sigma_t$ (MPa) |
|---|---|---|---|---|---|---|
| Soil | 1900 | 0.035 | 0.014 | 30.000 | 16.0 | 0.009 |
| Rock | 2700 | 24.000 | 13.000 | 120.000 | 45.0 | 2.000 |

The Mohr–Coulomb model was selected to describe the mechanical behavior of the inhomogeneous material. As shown in Figure 12, the confining pressure was set as 0.5 MPa, 1.0 MPa, and 1.5 MPa, respectively, applied on the opposite surfaces in the x-direction. The axial stress was applied by displacement on the top surface in the y-direction with the loading rate of $5.0 \times 10^{-4}$ mm/step. The y-directional displacement of the elements on the bottom surface and the z-directional deformation of all elements were restricted.

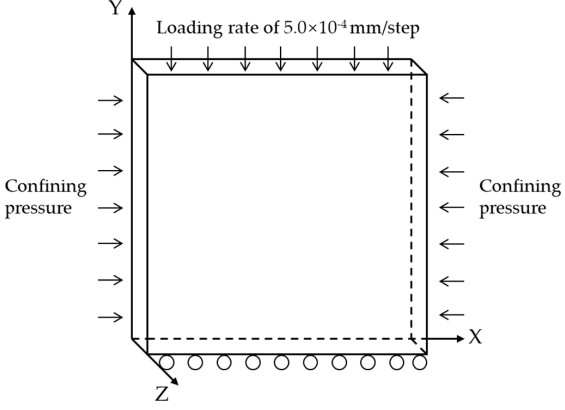

**Figure 12.** Boundary condition of samples in numerical tests.

## 4. Results

### 4.1. Strength of Samples with Different Sizes

By conducting numerical triaxial mechanical tests using Flac3D based on the finite difference method (FDM), a series of stress–strain curves of samples with different sizes were obtained (Figure 13). These curves tended to show a similar pattern when the size of the samples was large enough. Compressive strength is an important mechanical property of the SRM. The variability of the compressive strength $\sigma_c$ of the SRM decreases when the size of the sample increases.

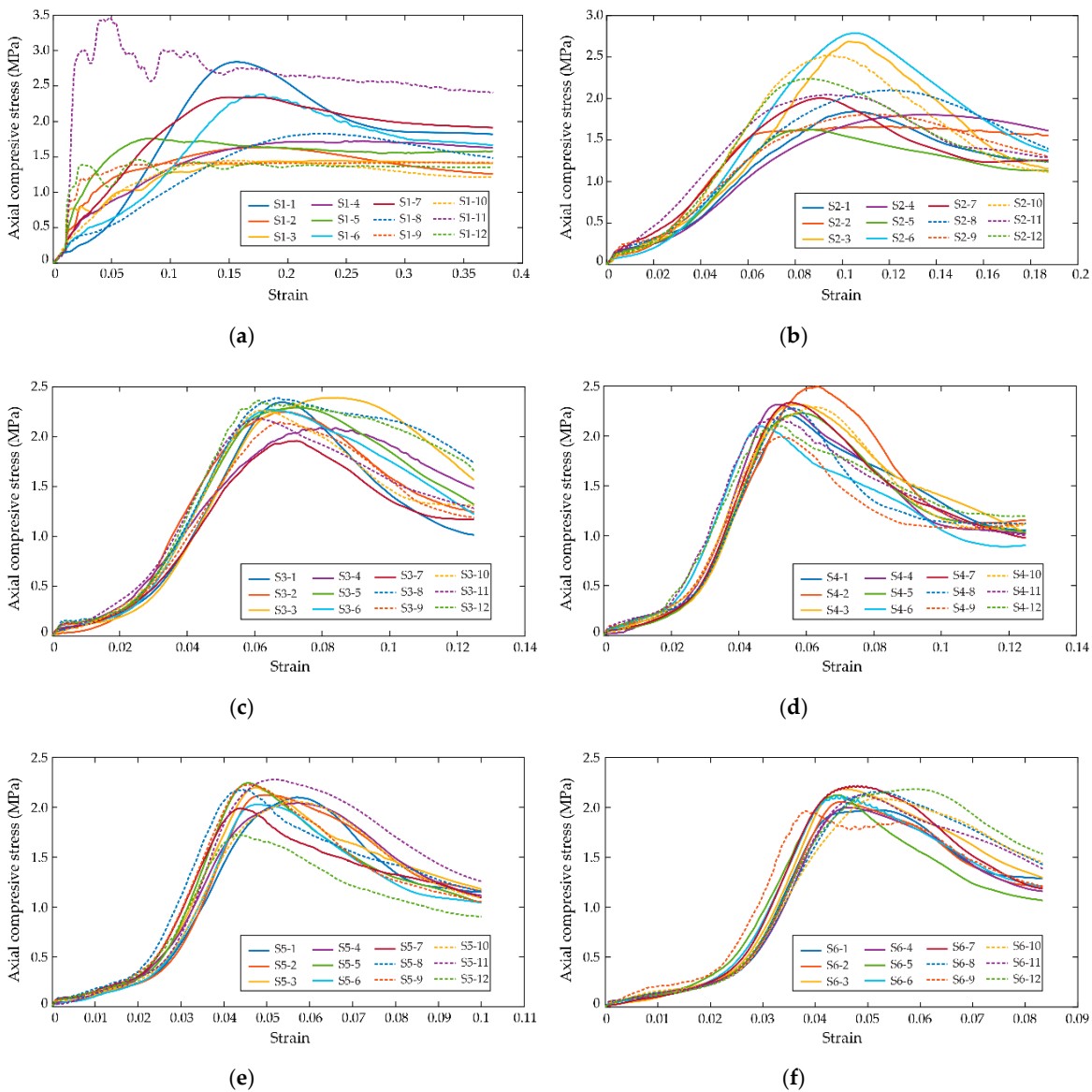

**Figure 13.** Stress–strain curves of samples with different sizes (confining pressure = 0.5 MPa): (**a**) $40 \times 40$ mm$^2$; (**b**) $80 \times 80$ mm$^2$; (**c**) $120 \times 120$ mm$^2$; (**d**) $160 \times 160$ mm$^2$; (**e**) $200 \times 200$ mm$^2$; (**f**) $240 \times 240$ mm$^2$.

As illustrated in Figure 14, the variability in the numerical experimental compressive strength $\sigma_c$ decreases as the size of the sample increases. The compressive strength $\sigma_c$ converges to a mean value of 2.10 MPa for the $240 \times 240$ mm$^2$ samples.

In addition to the compressive strength, the scale effect of samples in the shear resistant characteristic of the SRM was also investigated for its importance in engineering applications. Pseudo

triaxial compressive tests were conducted under confining pressures of 0.5 MPa, 1.0 MPa, and 1.5 MPa. The shear strength of the SRM can be estimated by linear regression between the compressive strength and confining pressure based on the Mohr–Coulomb strength theory:

$$\tau_f = c + \sigma \tan \varphi. \tag{4}$$

The r-square value, which is a measure for evaluating the goodness of fit, of each sample is larger than 0.9930, indicating good fit of the linear relationship. As a result, the cohesion $c$ and internal friction angle $\varphi$ can be obtained by the intercept on the stress axis and the slope from the linear regression.

For the shear resistance strength parameters shown in Figure 15, the significant effect of sample size on the cohesion $c$ was captured. When the length of the samples reached 240 mm, the values of cohesion converged to a mean value of 1.28 MPa. The mean cohesion of the samples increased as the length of the samples increased. On the other hand, the internal friction angle of the SRM did not show a noticeable variability for different sample sizes, even for the smallest sample size. The internal friction angle decreased slightly with the increasing of the sample size, approaching a mean value of 58.87°.

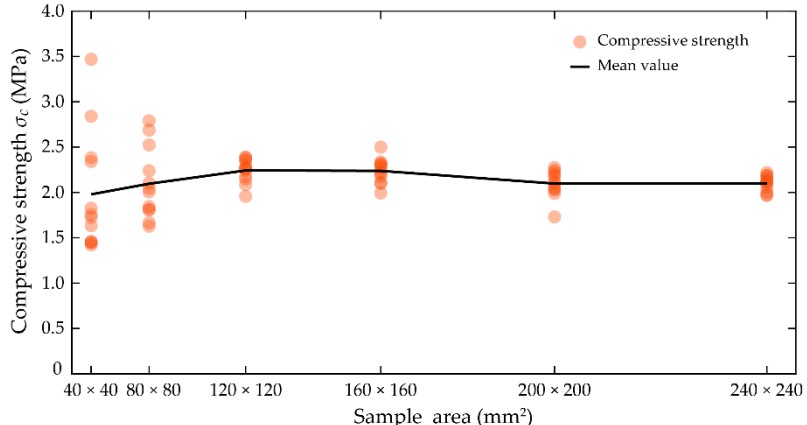

**Figure 14.** Compressive strength of different sizes (confining pressure = 0.5 MPa).

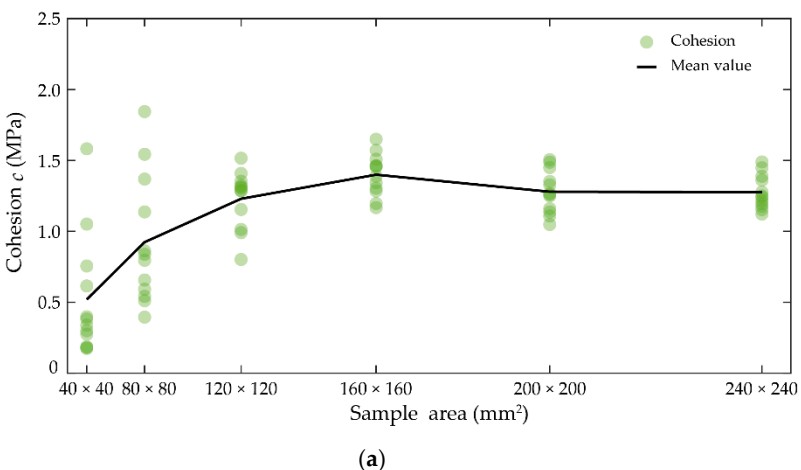

(**a**)

**Figure 15.** *Cont.*

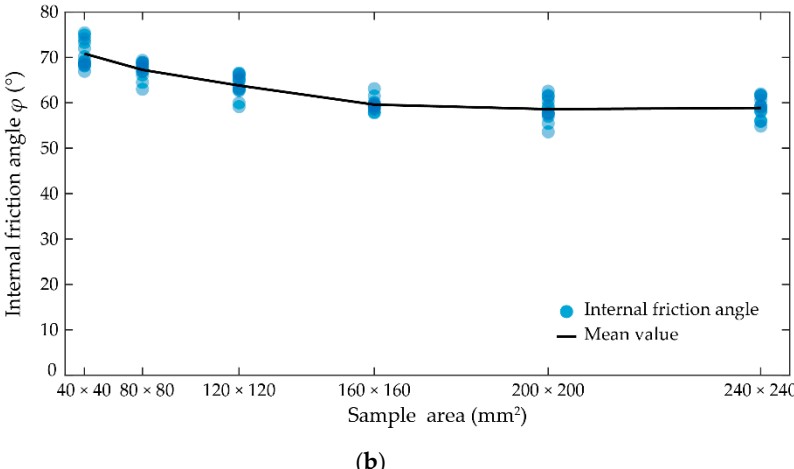

(**b**)

**Figure 15.** Strength parameters of the SRM changing with sample scale: (**a**) cohesion *c*; (**b**) internal friction angle $\varphi$.

## 4.2. RVE Estimation

In order to describe the variability of the mechanical parameters quantitatively, the coefficient of variation (CV) was chosen [46], which is given by the following:

$$\text{CV} = \frac{\text{SD}}{\mu}, \quad \text{SD} = \sqrt{\frac{1}{N-1}\sum_{i=1}^{N}(x_i - \mu)^2}, \quad \mu = \frac{1}{N}\sum_{i=1}^{N}x_i, \tag{5}$$

where $x_i$ is the strength parameter of the *i*-th sample; *N* is the number of samples and *N* = 12 in this study; and SD and $\mu$ stand for the standard deviation and mean value, respectively. The CV gives a quantitative description on the spatial dispersion of the data: the smaller the value of the CV, the lower the data dispersion or the closer the data to the mean value.

The CV of the compressive strength $\sigma_c$, cohesion *c*, and internal friction angle $\varphi$ are listed in Table 2. The changes in the CV of these macro mechanical properties of the SRM for different sample sizes are shown in Figure 16.

It is clearly seen that the CV of compressive strength and cohesion decreased with the increase of the sample size, especially when the length of the samples was less than 120 mm. The CV of cohesion was generally larger than that of compressive strength. The internal friction angle had the lowest CV. The behavior of the CV agrees with the observation on the mechanical properties of the SRM in Section 4.1. As a result, the CV is appropriate to be used in measuring the variability of mechanical properties of heterogeneous materials. If CV = 10% is taken as the acceptable convergence threshold of macro mechanical parameters, the corresponding size of samples of 240 × 240 mm² can be determined as the RVE in this study.

The changes in the CV of compressive strength with sample sizes were similar for different confining pressures (Figure 17). With the increase of sample size, the CV decreased rapidly at the beginning and fluctuated slightly when the sample size was larger than 120 × 120 mm². The lower the confining pressure, the larger the variability of compressive strength. The difference in the CV for different confining pressures decreased with the increase of the sample size.

**Table 2.** Coefficient of variation for strength parameters.

| Sample Size (mm$^2$) | Coefficient of Variation (%) | | |
| --- | --- | --- | --- |
| | Compressive Strength $\sigma_c$ | Cohesion $c$ | Internal Friction Angle $\varphi$ |
| $40 \times 40$ | 32.87 | 82.00 | 4.18 |
| $80 \times 80$ | 18.62 | 49.29 | 2.85 |
| $120 \times 120$ | 5.87 | 16.46 | 3.78 |
| $160 \times 160$ | 5.95 | 10.46 | 2.49 |
| $200 \times 200$ | 7.06 | 11.77 | 4.49 |
| $240 \times 240$ | 4.05 | 9.31 | 4.01 |

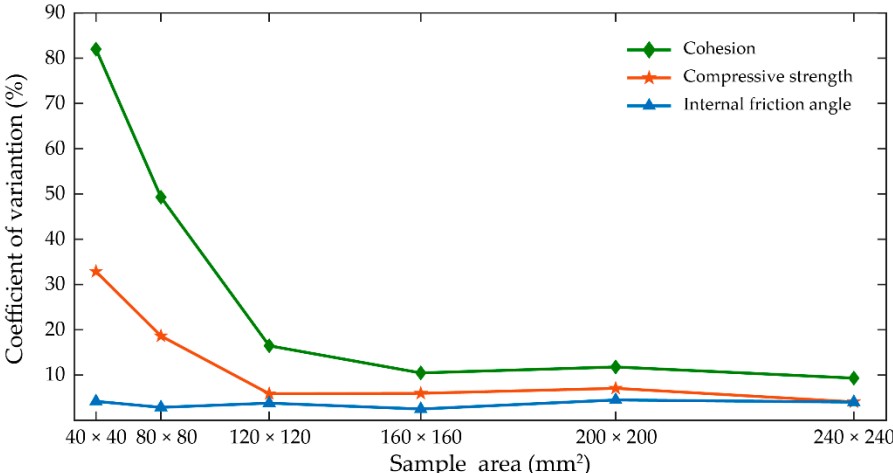

**Figure 16.** CV of compressive strength of the samples with different size.

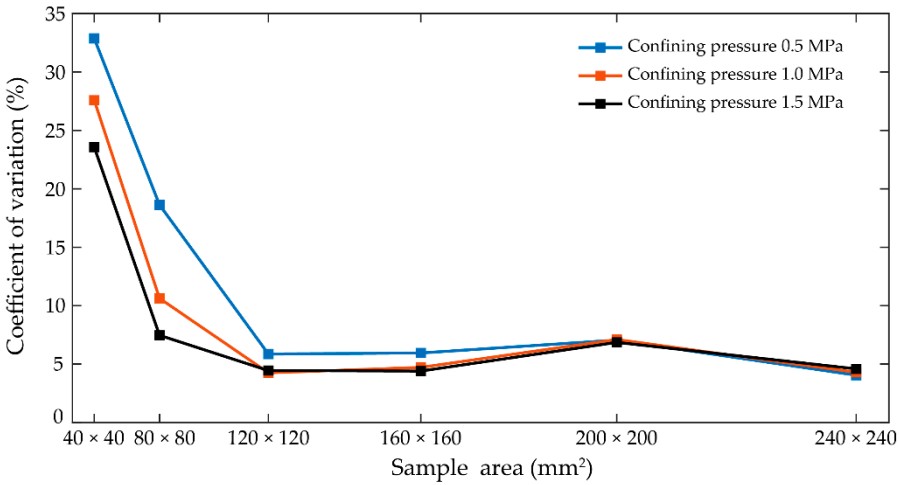

**Figure 17.** CV of compressive strength of the samples with different confining pressures.

## 5. Conclusions

This paper established a digital image processing (DIP)-based estimation method for the representative volume element (RVE) considering material heterogeneity. An efficient and simple connected-component labeling algorithm was introduced for geometry vectorization and microstructure identification. Numerical models were generated from the vectorized images for geomechanical numerical tests.

A color photo of soil and rock mixture (SRM) from a large landslide project was used as an example to illustrate the procedure. In order to investigate the size effect of the SRM, six sample sizes ranging from $40 \times 40$ mm$^2$ to $240 \times 240$ mm$^2$ were selected and twelve samples were extracted from the binarized image for each sample size. By conducting numerical triaxial testing of each sample, the variability of mechanical parameters, including compressive strength, cohesion, and internal friction angle, with different sample sizes was investigated. The following conclusions can be drawn:

(1) The variability of compressive strength and cohesion of the SRM decreases with the increase of the sample size, and reaches a stable small value when the sample size is large enough.

(2) The cohesion of the SRM shows a relatively larger variability than compressive strength and internal friction angle.

(3) With the increase of confining pressure, the variability of compressive strength decreases.

(4) By specifying an acceptable threshold for the coefficient of variation (CV), it can be used to estimate the RVE of the SRM. The DIP-based RVE estimation is a trade-off between obtaining accurate mechanical properties and reducing computational efforts in numerical tests.

## 6. Patents

Determination of the scale of representative volume element of soil–rock mixtures based on digital image processing technology, Patent number: ZL 201511019860.0 (authorized), China.

**Author Contributions:** Conceptualization, L.Y., Q.M., and W.X.; methodology and software, L.Y. and Q.M.; resources, W.X., W.-C.X., and H.W; formal analysis and writing—original draft preparation, L.Y. and M.S.; writing—review and editing, Q.M., W.-C.X., and M.S.; funding acquisition, W.X., H.W., and Q.M.

**Funding:** This research was financially supported by the National Key R & D Program of China, grant number 2018YFC1508501, the National Natural Science Foundation of China, grant number 11772118, the China Postdoctoral Science Foundation Funded Project, grant number 2017M610290, and the 111 Project.

**Conflicts of Interest:** The authors declare no conflict of interest.

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
