# Peer review of "Numerical Determination of RVE for Heterogeneous Geomaterials Based on Digital Image Processing Technology"

_processes, doi:10.3390/pr7060346_

Round 1
Reviewer 1 Report
Reference: processes-513276
Title: Numerical Determination of RVE for Heterogeneous Geomaterials Based on Digital Image Processing Technology
In this paper, “well known” digital image processing technology is used to estimate the representative volume element. Then performed the triaxial test and statistical manipulation to demonstrate the effect of sample size on the mechanical properties of different samples.
Following comments associated with the paper are presented:
· The authors also published the following paper
Qingxiang Meng, Huanling Wang, Weiya Xu, Qiang Zhang, (2018) "A coupling method incorporating digital image processing and discrete element method for modeling of geomaterials", Engineering Computations, Vol. 35 Issue: 1, pp.411-431, https://doi.org/10.1108/EC-11-2016-0390
Many equations and Figures are copied from the above reference without appropriate citation.
· In this manuscript, two aspects are discussed. First, the digital image processing. Second, Triaxial test. Both of them are well known. What is the originality of this paper?
· Equations 1-3 are the main features of this manuscript. Those equations are discussed in many textbooks and journals. Authors mentioned in the text that they proposed “digital image processing” for RVE. What is new in the proposal? What is the limitation of existing methods which motivated for new DIP?
· From Figure 9, it looks like after the boundary smoothing procedure, the boundary has changed, like Figures 9-a and 9-f. What is the impact of this change to the result?
· Geometry vectorization of binary image to Gmsh’s *.GEO and *.MSH files are not clear. Further explanations are essential here.
· From the triaxial test, it is well known that sample, size, shape, particle size distribution, loading condition, sample conditions have a significant effect. Such a numerical technique in FEM/FDM are very common. From *.MSH file it is not clear the boundary conditions and the initial conditions. What types of triaxial test authors performed? 2D or 3D?. From DIP and after manipulation, the procedure to obtain the 2D image in Figure is clear, but how did authors obtain Figure 11-b, from 2D. Besides, authors did not use the citation for Gmsh2Flac interface tool. Additionally, citation 48 is not appropriate. It is recommended to use textbook reference.
· Authors did not discuss anything about the saturation state of the sample.
In my opinion, the authors did not present the originality of the work. Also, the literature review and the justification of the problem statement are also not adequate. Therefore, the manuscript requires major revision.
Author Response
The authors would like to express their sincere thanks for the valuable comments from the editor and the two reviewers, which have greatly improved this manuscript. According to the comments and suggestions of the editor and the reviewers, a major revision is made for our manuscript, the contents of which are detailed below.
Point 1: In this paper, “well known” digital image processing technology is used to estimate the representative volume element. Then performed the triaxial test and statistical manipulation to demonstrate the effect of sample size on the mechanical properties of different samples.
Following comments associated with the paper are presented:
The authors also published the following paper
Qingxiang Meng, Huanling Wang, Weiya Xu, Qiang Zhang, (2018) "A coupling method incorporating digital image processing and discrete element method for modeling of geomaterials", Engineering Computations, Vol. 35 Issue: 1, pp.411-431,
https://doi.org/10.1108/EC-11-2016-0390
Many equations and Figures are copied from the above reference without appropriate citation.
Response 1: Citation of equation (2) and (3), and Figure 3 have been added accordingly.
Point 2: In this manuscript, two aspects are discussed. First, the digital image processing. Second, Triaxial test. Both of them are well known. What is the originality of this paper?
Response 2: It has to be clarified that the main purpose of this paper is providing a representative volume element (RVE) size estimation method for heterogeneous geomaterial on the basis of digital image processing (DIP) procedure. To the best knowledge of the authors, this paper is the first one for DIP-based RVE estimation method for heterogeneous geomaterial.
Point 3: Equations 1-3 are the main features of this manuscript. Those equations are discussed in many textbooks and journals. Authors mentioned in the text that they proposed “digital image processing” for RVE. What is new in the proposal? What is the limitation of existing methods which motivated for new DIP?
Response 3.1: The RVE size determination is of critical significance in the numerical modelling and simulation of heterogeneous material.
Response 3.2: Existing method mainly based on the laboratpry test, which consumes a large amount of money and labor. Although the DIP method is widely used in various fields, the merit of it has not been highlighted in the RVE estimation of heterogeneous geomaterial.
Point 4: From Figure 9, it looks like after the boundary smoothing procedure, the boundary has changed, like Figures 9-a and 9-f. What is the impact of this change to the result?
Response 4: It is known that, a digital image consists of a rectangular matrix of pixels. When the pre-processing procedure is finished, the jagged edge can be detected by microstructure detection method. However, the jagged edge cannot reflect the real edge of the rock. Furthermore, an excessive number of nodes will negatively affect the generation of computational models and the stress distribution during numerical simulation. The boundary processing procedure intends to reduce the number of nodes and smooth the boundaries. The smoothed polygon is acceptable when a low threshold is set.
As a result, smoothed boundaries which have much less nodes are more convenient for establishing numerical models, can represent the real rocks edge, and mechanically perform closer to real rocks.
Point 5: Geometry vectorization of binary image to Gmsh’s *.GEO and *.MSH files are not clear. Further explanations are essential here.
Response 5: Thank you so much for your suggestion. The data format of the input geometry file *.GEO and the output mesh file *.MSH are essential for the application of Gmsh. *.GEO file consists of basic information including a predefined mesh size, elementary entities (points, lines, surfaces), physical entities (points, lines, surfaces). *.MSH file is used to store meshes and associated post-processing datasets.
Detailed explanation is complemented in Page 10 from lines 235 to 243.
“For the convenience of later analysis, Gmsh software, a fast, light, and user-friendly interactive mesh tool, is employed to generate mesh grid. As a result, the geometry information including mesh size, node coordinates, line segments, and line loops should be included in a *.GEO file, which is the recommended input file format. A program is developed to write the geometry information in the format of *.GEO file.
By importing the *.GEO files of SRM samples obtained in Section 2.2 into Gmsh, 2D finite element meshes are generated easily. A mesh example for a sample of size 40×40 mm2 is shown in Figure 11a, of which the meshes is stored in *.MSH file format containing the physical group names (“rock” and “soil”), the nodes, and the elements (the element type of 3-node triangle is used for all samples).”
Point 6: From the triaxial test, it is well known that sample, size, shape, particle size distribution, loading condition, sample conditions have a significant effect. Such a numerical technique in FEM/FDM are very common. From *.MSH file it is not clear the boundary conditions and the initial conditions. What types of triaxial test authors performed? 2D or 3D? From DIP and after manipulation, the procedure to obtain the 2D image in Figure is clear, but how did authors obtain Figure 11-b, from 2D. Besides, authors did not use the citation for Gmsh2Flac interface tool. Additionally, citation 48 is not appropriate. It is recommended to use textbook reference.
Response 6.1: As illustrated in previous paragraph, *.MSH is a file storing the 2D meshes generated from Gmsh. For later numerical simulation, the 2D meshes have to be transformed into FDM acceptable entity grid model, i.e., 3D FDM numerical model stored in *.FLAC3D file in this study. As a result, an interface program Gmsh2Flac is developed to transform 2D meshes into 3D entity models.
By reading the 2D mesh information including points coordinates, elements, and physical groups, mapping the 2D meshes (3-node triangle plane elements) to the 3D meshes (6-node wedge elements) along z-direction with a certain thickness, and storing the 3D meshes information in *.FLAC3D format, pseudo three-dimensional numerical models are obtained.
Detailed explanation is added into the revised manuscript in Page 11 from lines 247 to 252.
“In later numerical study, a finite difference method (FDM) based software Flac3D is applied to carry out numerical triaxial mechanical tests of the SRM for estimating the size of RVE. An interface program Gmsh2Flac is written to map the 2D meshes (3-node triangle elements) to the 3D meshes (6-node wedge elements) along z-direction with a certain thickness (Figure 11b). The 3D meshes are stored in *.FLAC3D file which is acceptable as numerical sample in Flac3D. As a result, 72 pseudo three-dimensional finite element models are obtained.”
Response 6.2: Considering the pseudo three-dimensional numerical model, the pseudo triaxial compressive tests are conducted correspondingly. A figure illustrating the boundary conditions and loading direction on the 3D *.FLAC3D model is complemented in Figure 12 in Page 12.
“As shown in Figure 12, the confining pressure is set as 0.5 MPa, 1.0 MPa, and 1.5 MPa, respectively, applied on the opposite surfaces in the x-direction. The axial stress was applied by displacement on the top surface in y-direction with the loading rate of 5.0×10-4 mm/step. Y-directional displacement of the elements in bottom surface and z-directional deformation of all elements are restricted.” (Page 11, lines 261-265)
Response 6.3: The citation 48 has been replaced by a textbook as you suggested.
Lu, T. Soil Mechanics, 2 ed.; Hohai University Press: Nanjing, 2005. ISBN: 9787563016457
Point 7: Authors did not discuss anything about the saturation state of the sample.
Response 7: Yes, the saturation status is not mentioned. However, this paper focuses on proposing an effective RVE determination method of heterogeneity geomaterials.

Reviewer 2 Report
In the paper the material heterogeneity is analysed with the use of the numerical digital image processing method. The method of the analysis as well as the description of the results are discussed in quite good manner with one exception given below. The size of the RVE cannot be characterized by one linear parameter and in my opinion should be replaced by the area of the RVE especially in view of the estimation of the scale effects (e.g. Fig. 15). I am awiating that the authors wiill explain the above problem in the revised version of the paper.
Author Response
The authors would like to express their sincere thanks for the valuable comments from the editor and the two reviewers, which have greatly improved this manuscript. According to the comments and suggestions of the editor and the reviewers, a major revision is made for our manuscript, the contents of which are detailed below.
Point 1: In the paper the material heterogeneity is analysed with the use of the numerical digital image processing method. The method of the analysis as well as the description of the results are discussed in quite good manner with one exception given below. The size of the RVE cannot be characterized by one linear parameter and in my opinion should be replaced by the area of the RVE especially in view of the estimation of the scale effects (e.g. Fig. 15). I am awaiting that the authors will explain the above problem in the revised version of the paper.
Response 1: The figures from Fig. 14 to Fig. 17 which describe the scale effect of strength parameters have been replotted. The axis of side length is replaced by the sample area as you suggested referring to the revised manuscript please.

Round 2
Reviewer 1 Report
Accepted in the present form.
Reviewer 2 Report
I accept it but I am not completely satisfied.